# Filling the Image Information Gap for VQA:
# Prompting Large Language Models to Proactively Ask Questions

**Ziyue Wang[1], Chi Chen[1], Peng Li[✉2,3], Yang Liu[✉1,2,3]**

[1]Department of Computer Science and Technology, Tsinghua University, Beijing, China
[2]Institute for AI Industry Research (AIR), Tsinghua University, Beijing, China
[3]Shanghai Artificial Intelligence Laboratory, Shanghai, China
{wangziyue22,chenchi19}@mails.tsinghua.edu.cn
lipeng@air.tsinghua.edu.cn
liuyang2011@tsinghua.edu.cn

## Abstract

Large Language Models (LLMs) demonstrate impressive reasoning ability and the maintenance of world knowledge not only in natural language tasks, but also in some vision-language tasks such as open-domain knowledge-based visual question answering (OK-VQA). As images are invisible to LLMs, researchers convert images to text to engage LLMs into the visual question reasoning procedure. This leads to discrepancies between images and their textual representations presented to LLMs, which consequently impedes final reasoning performance. To fill the information gap and better leverage the reasoning capability, we design a framework that enables LLMs to proactively ask relevant questions to unveil more details in the image, along with filters for refining the generated information. We validate our idea on OK-VQA and A-OKVQA. Our method continuously boosts the performance of baselines methods by an average gain of 2.15% on OK-VQA, and achieves consistent improvements across different LLMs[1].

## 1 Introduction

Large-scale language models (LLMs) have exhibited impressive capabilities in terms of their world knowledge and reasoning abilities, leading to remarkable achievements in various Natural Language Processing (NLP) tasks such as commonsense reasoning (Tamborrino et al., 2020; Wei et al., 2022) and open-domain question answering (Li et al., 2022; Kamalloo et al., 2023). Building upon the success in the realm of text, recent research has explored the utilization of pre-trained LLMs in Vision-Language (VL) tasks. These studies have shown promising performance, especially for knowledge-intensive tasks such as knowledge-based Visual Question Answering (VQA) (Marino

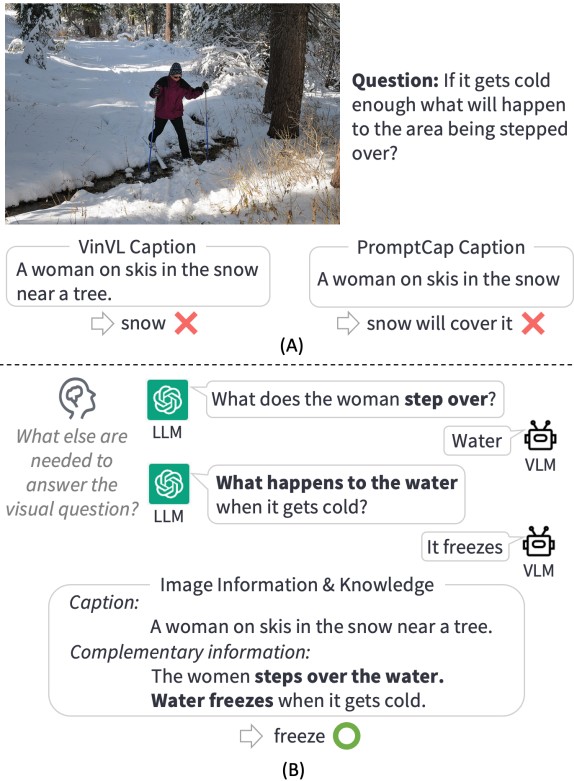

Figure 1: An example of our framework (B) compared to baselines (A). In the two methods in (A), the caption models do not provide precise information of "what is being stepped over", resulting in hallucinated answers. Our method (B) empowers the LLM to actively seek and acquire missing information by querying the VLM.

et al., 2019; Schwenk et al., 2022), where both image understanding and external knowledge are imperative for answering open-ended questions.

The key challenge of leveraging LLMs in VL tasks is to bridge the gap between images and text, *i.e.*, enabling LLMs to understand images. To address this challenge, two approaches have been investigated. The first approach extends the LLM with visual perception modules to form a Vision-Language Pre-training (VLP) model (Alayrac et al., 2022; Chen et al., 2022; Li et al., 2023). Despite the high performance, this approach requires training on large-scale image-text pairs, which can be com-

---

✉Corresponding authors: Peng Li and Yang Liu.
[1]Code will be released at https://github.com/THUNLP-MT/FIIG.

putationally expensive, particularly for LLMs with massive parameters such as GPT-3 (Brown et al., 2020) and PaLM (Chowdhery et al., 2022). The second approach transforms images into textual representations, which are then used as prompts for the LLMs (Yang et al., 2022; Hu et al., 2023a; Chen et al., 2023; Guo et al., 2023). This training-free approach is more cost-effective and enables LLMs to be utilized in a more flexible paradigm, allowing for easy adaptation to different tasks.

However, as discussed in the works of Yang et al. (2022) and Hu et al. (2023a), general image captions may lack the subtle information required to answer visual questions. To resolve this, Yang et al. (2022) compromise captions with image tags, and Hu et al. (2023a) propose incorporating questions into the caption generation process. Despite their successes, it remains impractical to reveal every subtle detail necessary to answer visual questions in a single concise description. As illustrated in Figure 1, the captions fail to spot the "area being stepped over" as the "water", resulting in hallucinated answers. Two primary concerns exist regarding existing image-to-text conversion approaches for VQA: (1) The converted textual descriptions might be insufficient to solve the visual questions, or could contain misleading content; (2) Existing methods convert images to text as a preprocessing step of the input. This one-off conversion is a lossy compression of the conveyed information, and does fully provoke the reasoning ability of LLMs.

In this paper, we present a framework where LLMs proactively interact with Vision-Language Models (VLMs) to gather specific information of interest, as depicted in Figure 1. This interaction is aimed at automatically seeking and regaining details that may be omitted during the image-to-text conversion. To enhance the informativeness of the generated questions and the correctness of their corresponding answers, we design a refinement module to summarize only the useful information for generating the final answers. We validate our approach on OK-VQA and A-OKVQA datasets and conduct experiments across different LLMs. Our contributions are as follows:

- We design a model agnostic framework that allows LLMs to proactively interact with VLMs to unveil missing information.

- Our method can rectify inaccurate information generated during the image-to-text transformation process and minimize the ambiguity of

the converted textual information.

- We achieve an average gain of 2.15% on OK-VQA over baselines, and attain consistent improvements across different LLMs.

# 2 Related Work

## 2.1 In-Context Learning for OK-VQA

In-context learning refers to the ability of LLMs to adapt to new tasks given only a few examples without parameter tuning (Brown et al., 2020). To exploit the knowledge possessed by LLMs and extend this ability to OK-VQA, PICa (Yang et al., 2022) converts images into captions and tags, and applies in-context learning with GPT-3. A following work, PromptCap (Hu et al., 2023a), fine-tunes a question-guided captioning model, urging the target messages to appear in the captions. Instead of describing an image by a single caption, Chen et al. (2023) represent an image as the combination of its regions in a Chain-of-Thought (CoT) style (Wei et al., 2022). Prophet (Shao et al., 2023) argues that indistinct captions lead to aimless predicting, and proposes to provide answer candidates with corresponding confident scores as references. These methods select in-context examples according to the similarities between training and test instances.

However, there are unexpected information loss during the conversion from images to text. These methods conduct a compressive one-time conversion to turn images into text, while we prompt the LLM to iteratively ask for detailed information. Our method is orthogonal to these approaches and can continually improve their performances.

## 2.2 New Question Generation for VQA

In VQA tasks, some questions are ambiguous and might have different answers (Bhattacharya et al., 2019). Uehara et al. (2022) propose to generate new questions to assist the reasoning of the original questions. They train a visual question generator with supervision from human annotated dataset (Selvaraju et al., 2020). It is evaluated on VQA dataset (Antol et al., 2015), and is not extensible to open-domain knowledge-based VQA. Img2prompt (Guo et al., 2023) describes a zero-shot approach on OK-VQA by generating an extra new question for each instances to imitate the few-shot setting without actually using multiple in-context examples. Its question generation procedure is irrelevant to the images. Instead of depending on human annotations, given the question and

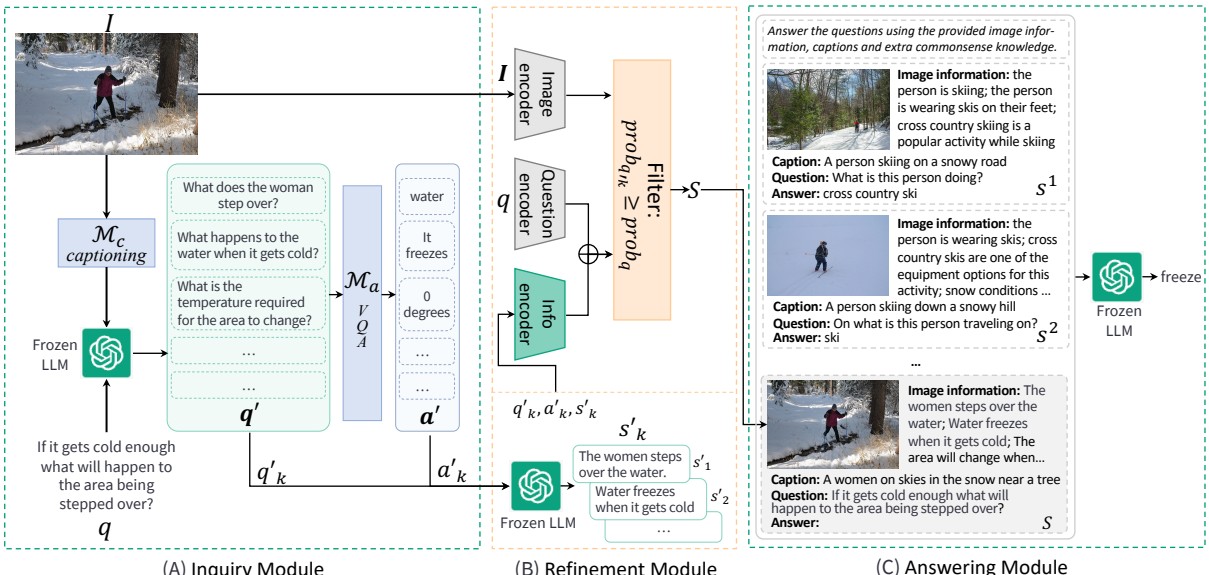

Figure 2: Our proposed framework consists of three modules. First, in the **inquiry** module (§3.1), we prompt the LLM to generate new questions for the missing image information required to answer the original question, and obtain answers from a VLM. Then, a **refinement** module (§3.2) is adopted to summarize the questions and answers, filtering and extracting useful information from them. Finally, in the **answering** module (§3.3), the LLM is prompted to predict the final answer with the augmented image information.

image information, we directly prompt LLMs to generate new questions targeting the missing information. This is not constrained on pre-annotated datasets and allows us to uncover more potential information and knowledge.

## 2.3 Incorporating LLMs and VLMs for Vision-Language Tasks

Some concurrent works have also incorporated LLMs and VLMs for VL tasks. ChatCaptioner (Zhu et al., 2023), for instance, aims to capture richer details from images by progressively generating questions with an LLM and then answering these questions using a VLM. The resulting answers are summarized to form image descriptions. However, this approach places a stronger emphasis on the quantity of detected image details rather than their correctness. In some VL tasks, such as VQA, this inevitably introduces noise, leading to inaccurate image descriptions that may result in incorrect model predictions. AVIS (Hu et al., 2023b) also explores the decomposition of visual questions, but its primary focus lies in the planning procedures for tool-use and their corresponding executions.

## 3 Method

Our overall framework, as illustrated in Figure 2, comprises three key modules: **inquiry**, **refinement** and **answering**. Given the image caption and the

question, the inquiry module prompts an LLM to generate new questions that seek for missing image information required to answer the original question, and uses a VLM to answer them based on the image (§3.1). Then, we adopt a refinement module to summarize information from the generated questions and answers, filtering and extracting the relevant and useful information from them (§3.2). Finally, the answering module prompts the LLM to predict the final answer to the original question with the augmented image information (§3.3).

Before delving into the details of our method, we shall declare some important notations. We use $I$, $c$, $q$ to denote the image, its caption, and the original question, respectively. The caption $c$ is generated by the image captioning model $\mathcal{M}_c$:

$$c = \mathcal{M}_c(I), \qquad (1)$$

which is used as the preliminary image information presented to LLMs for question generation. The generated new questions are denoted as $\boldsymbol{q}' = [q'_1; q'_2; \ldots; q'_K]$, with the corresponding answers as $\boldsymbol{a}' = [a'_1; a'_2; \ldots; a'_K]$, where $K$ represents the total number of questions.

## 3.1 Prompting LLM to Proactively Ask for Information

We leverage the reasoning capabilities of the LLM to identify the essential image information that may

be lost during the image-to-text conversion process. Specifically, given the image caption $c$ and the original question $q$, we first prompt the LLM to generate $K$ new questions $\boldsymbol{q}'$ that inquire about the additional image information necessary to answer $q$. Suppose that the $k$-th question of $\boldsymbol{q}'$ has $L$ tokens, denoted as $q'_k = (y_k^1, y_k^2, \ldots, y_k^L)$, the decoding process can be formulated as:

$$y_k^l = \arg\max_{\hat{y}_k^l} p_{\text{LLM}}\left(\hat{y}_k^l \middle| y_k^{<l}; \boldsymbol{p_q}, q, c\right), \quad (2)$$

where $\boldsymbol{p_q}$ is the instruction prompt. The outline of the prompt $\boldsymbol{p_q}$ for LLM is as follows:

```
/* Instruction for the decomposition task */
Please decompose the TARGET-QUESTION into K
sub questions:
/* n in-context examples */
TARGET-QUESTION: q^1 \n Catpion: c^1
Sub questions: 1. q'^1_1, 2. q'^1_2, ...
......
TARGET-QUESTION: q \n Caption: c
Sub questions:
```

Then, we employ a visual question answering model $\mathcal{M}_a$ to answer these new questions based on the original image as:

$$a'_k = \mathcal{M}_a(q'_k, I), \quad (3)$$

where $a'_k$ refers to the answer to the $k$-th question.

To better understand the role of the generated questions and answers, we conducted a preliminary experimental analysis. Specifically, we concatenate each question $q'_k$ to its answer $a'_k$ to form a QA pair $q'_k a'_k = [q'_k; a'_k]$, and prompt the LLM to answer OK-VQA questions given this representation. We investigate the results of prompting LLM with different contexts: 1) the original question $q$ only; 2) all the QA pairs; 3) one randomly selected QA pair and 4) the best-performing QA pair. For the last case, for each question we calculate the accuracy scores when prompting with each QA pair, and then select the maximum score among them, which represents an upper bound on the performance. These accuracy scores are calculated by the soft scores following Goyal et al. (2017).

From the results presented in Table 1, we can draw two key conclusions. First, the generated questions and answers indeed contain information that helps to answer the original question, comparing the results of *Best* and *Original*. Second, the generated QA pairs are noisy, as neither using all QA pairs nor randomly selecting one improves the performance. This highlights the necessity of an information refinement process to filter out irrelevant or misleading information in the generated pairs.

| Selection | Original | All | Random | Best |
|---|---|---|---|---|
| Accuracy | 45.17 | 45.25 | 41.31 | 63.52 |

Table 1: Preliminary experiments on the effects of the generated QA pairs. We report the average accuracy scores of prompting the LLM to answer each OK-VQA question with: (1) "Original": the original question $q$ only; (2) "ALL": all the QA pairs; (3) "Random": one randomly selected QA pair. "Best" refers to the best-performing QA pair (*i.e.*, the upper bound).

## 3.2 Refining the Generated Information

Inspired by the preliminary experiments in §3.1, we design a refinement module that can extract useful information for the LLM to answer the original question from the noisy generated QA pairs. Our refinement module includes a summarization stage and a filtering stage.

Firstly, we summarize $q'_k$ and the corresponding answer $a'_k$ into a narrative description $s'_k$. Denoting $s'_k$ as a $L$-token target sequence, we have:

$$s_k'^l = \arg\max_{\hat{s}_k'^l} p_{\text{LLM}}\left(\hat{s}_k'^l \middle| s_k'^{<l}; \boldsymbol{p_s}, q'_k, a'_k\right), \quad (4)$$

where $l$ is the $l$-th token of the summary $s'_k$ and $\boldsymbol{p_s}$ is the instruction prompt. The complete templates used in this section are listed in Appendix F.

Then, we apply a filter to assess the helpfulness of summary $s'_k$ to the final prediction. The output is a contribution score of $s'_k$ given the image $I$, the question $q$, and different combinations of text inputs including question $q'_k$, answer $a'_k$, summary $s'_k$ and question-answer pair $[q'_k; a'_k]$.

Specifically, our refinement module consists of two types of encoders, text encoder $\text{Enc}_t$ and image encoder $\text{Enc}_v$, and a 3-layer multilayer perceptron (MLP). We use the pre-trained weights of CLIP (Radford et al., 2021) to initialize our text and image encoders. Given $I$, $q$ and $s'_k$, we first generate the visual features $\boldsymbol{h}_{visual}$ and the textual features $\boldsymbol{h}_{text}^k$ as:

$$\boldsymbol{h}_{visual} = \text{Enc}_v(I), \quad (5)$$

$$\boldsymbol{h}_t = \text{Enc}_t(t), t = \{q, q'_k, a'_k, q'_k a'_k, s'_k\}, \quad (6)$$

$$\boldsymbol{h}_{text}^k = \text{Avg}(\boldsymbol{h}_{t=\{q'_k, a'_k, q'_k a'_k, s'_k\}}, \boldsymbol{h}_{t=q}), \quad (7)$$

where $\boldsymbol{h}_{text}^k$ is the average of the features of each kind of textual inputs. Then we calculate the fused features of the image $I$ and the summary $s'_k$ as:

$$\boldsymbol{z}^k = \text{MLP}([\boldsymbol{h}_{text}^k; \boldsymbol{h}_{visual}]). \quad (8)$$

| Method | LLM | Image Information | OK-VQA | A-OKVQA |
|---|---|---|---|---|
| | | *Results with accessible LLMs* | | |
| PICa | `davinci` | Captions (G) + Tags | 48.0* | - |
| PICa† | `text-davinci-002` | Captions (G) + Tags | 49.67 | 49.18 |
| + Ours | | Captions (G) + Tags + refined details | 53.76 +4.09 | 51.13 +1.95 |
| PromptCap† | `text-davinci-002` | Captions (Q) | 53.50 | 52.99 |
| + Ours | | Captions (Q) + refined details | 54.42 +0.92 | 53.21 +0.22 |
| Prophet | `text-davinci-002` | Caption (G) + Candidates | 57.91 | 58.2* |
| + Ours | | Caption (G) + Candidates + refined details | 59.34 +1.43 | **59.79*** +1.59 |
| | | *Results with unavailable LLMs* | | |
| PromptCap | `code-davinci-002` | Captions (Q) | 60.4 | 56.3 |

Table 2: Direct answer accuracy on OK-VQA (test) and A-OKVQA (val). We use † to denote our reproduced versions, because PromptCap and PICa use different GPT-3 engines in their paper. PICa uses `davinci`; PromptCap uses `code-davinci-002`, which is deprecated. * refers to multi-query ensemble because the single query results of Prophet is not reported by Shao et al. (2023). The Captions (G) refers to the caption generated by general-purpose VLP, such as VinVL. The Captions (Q) refers to the PromptCap caption. Candidates refers to answers candidates predicted by VQA model. Refined details refer to our regained information.

To optimize the filter, we directly use the ground-truth VQA accuracy $y_{z^k}$ of each training instance $(I, q, s'k, y_{z^k})$ in OK-VQA as an intermediate supervision signal. The final loss is formulated as:

$$\mathcal{L} = -[y_{z^k} \log(p_{z^k}) + (1 - y_{z^k}) \log(1 - p_{z^k})], \quad (9)$$

where $p_{z^k} = \sigma(z^k)$ is the contribution score of $s'_k$ for answering $q$ given $I$. Please refer to Appendix A for detailed data construction process.

During inference, we exploit the refinement module to generate the refined information $S$. We first calculate the contribution score of the original question $q$ as $p_{z^q}$ with the trained filter. Then we select all the summaries $s'_k$ that have larger contribution scores $p_{z^k}$ than $p_{z^q}$ to form the refined information set $S$ as:

$$S = \{p_{z^k} | p_{z^k} \geqslant p_{z^q}\}_{k=1,2,3,\ldots,K}. \quad (10)$$

### 3.3 Answer Reasoning

The answering module prompts a frozen LLM to predict the final answer with both the converted image information such as the caption $c$ and our regained $S$. For the prompt template, we mostly follow PICa (Yang et al., 2022) as the other methods do, but add a new section for our refined image information as follows (denoted as *Template-Few-shot*):

```
/* Template-Few-shot */
Image information: S^n
Caption: C^n\n Question: q^n\n Answer: a^n
```

where $a^n$ is the ground-truth answer of question $q^n$ and $n$ refers to the number of shots used for in-context learning. We denote the template for the test data as *Template-Query*, which is as follows:

```
/* Template-Query */
Image information: S
Caption: C\n Question: q\n Answer:
```

Please refer to Appendix F for the complete prompt templates with instructions.

## 4 Experiments

We validate our method on open-domain knowledge-base VQA, and conduct experiments on OK-VQA and A-OKVQA. Implementation details are described in the following sections.

### 4.1 Dataset

**OK-VQA** (Marino et al., 2019) is an open-domain knowledge-based VQA dataset with about 9k training and 5k testing questions. The images come from MSCOCO (Lin et al., 2014). The annotated questions are all open-ended, and each of them is associated with 10 groundtruth answers. Due to the deprecation of dataset v1.0, we use v1.1 in this paper. **A-OKVQA** (Schwenk et al., 2022) augments OK-VQA by the scale, tasks and extra rationales. This dataset has are three splits, training (17k), validation (1.4K) and test (6.7k). A-OKVQA includes two tasks, direct answer (DA) and multiple choices (MC). The DA tasks is the same as OK-VQA, which requires to answer open-ended

questions with external knowledge. While the MC task asks to choose an answer from a close set with 4 choices. We focus on the open-domain setting and evaluate our method OK-VQA and the DA task of A-OKVQA. Both datasets employ the soft accuracy (Antol et al., 2015) as the evaluation metric.

## 4.2 Experimental Settings

We generate 3 new questions for each $q$, and apply the ensemble filters to refine the generated information. The ensemble filters contains all 4 types of inputs, $q'_k$, $a'_k$, $s'_k$ and $[q'_k; a'_k]$.

**Baseline methods.** We apply our method upon existing approaches of the same in-context learning paradigm, including PICa (Yang et al., 2022), PromptCap (Hu et al., 2023a) and Prophet (Shao et al., 2023). The image information of **PICa** comes from captions and tags (Microsoft Azure tagging API[2]). **PromptCap** follows similar settings, but replaces the captioning model by its fine-tuned question-aware caption model. Apart from the captions, **Prophet** supplies LLM with extra answer candidates obtained from their fine-tuned VQA models.

The best results of PICa and Prophet is reported on the 16-shot and 20-shot settings, respectively. They employ multi-query ensemble to further enhance the performance, where they prompt the LLM for 5 times with different in-context examples. We implement our method upon these baselines following the same settings, where the number of shot is 16 for PICa+Ours and PromptCap+Ours, and 20 for Prophet+Ours.

**LLMs.** For answer reasoning, the three baselines employ different LLMs. Prophet employs the LLM engine `text-davinci-002`, which is an InstructGPT model (Ouyang et al., 2022). PICa uses `davinci`, a GPT-3 model (Brown et al., 2020). PromptCap uses `code-davinci-002`, but is now deprecated and the authors suggest to replace it by `text-davinci-002` in the published code and model[3]. Considering the accessibility of LLMs and for fair comparisons, we employ the LLM engine used in Prophet (`text-davinci-002`) and reproduce the other two using the same engine. The `gpt-3.5-turbo-0301` engine is employed for question generation and summarization.

---

[2]Azure Computer Vision API v3.2 for tagging: https://westus.dev.cognitive.microsoft.com/docs/services/computer-vision-v3-2

[3]https://github.com/Yushi-Hu/PromptCap

| Method | OK-VQA | A-OKVQA |
|---|---|---|
| *Multimodal tuning* | | |
| LXMERT | - | 30.7 |
| BLIP-2 (FlanT5-XL) | 40.7 | - |
| VLC-BERT | 43.1 | 45.0 |
| GPV-2 | - | 48.6 |
| Flamingo (80B) | 57.8 | - |
| InstructBLIP (Vicuna-7B) | 62.1 | 64.0 |
| PaLM-E (562B) | 66.1 | - |
| *Methods querying external KBs (w/o LLMs)* | | |
| KRISP | 38.9 | 33.7 |
| RA-VQA | 54.5 | - |
| REVEAL | 59.1 | 52.2 |
| *Methods with LLMs* | | |
| Img2Prompt175B | 45.6 | 42.9 |
| PICa-Full | 48.0 | - |
| TRiG | 50.5 | - |
| KAT (ensemble) | 54.4 | - |
| REVIVE | 58.0 | - |
| assistGPT | - | 44.3 |
| PromptCap | 60.4 | 59.6 |
| Prophet | 57.9 | - |
| + Ours | 59.3 +1.4 | 57.9 |
| Prophet (ensemble) | 61.1 | 58.2 |
| + Ours (ensemble) | **61.3** +0.2 | **59.8** +1.59 |

Table 3: Comparisons to the previous methods of on OK-VQA (test) and A-OKVQA (val). The highest accuracy of methods using LLMs are **bolded**. The sota of the other two scopes are underlined. Please refer to Appendix C for the full results.

**VLMs.** We use BLIP-2 (Li et al., 2023) to predict answer $a'_k$ for $q'_k$. The choice of VLMs for obtaining caption $C$ varies alone baselines. Specifically, PICa and Prophet uses VinVL (Zhang et al., 2021) to convert images into captions, and PromptCap fine-tunes a question-aware caption model based on OFA (Wang et al., 2022).

## 4.3 Results

Table 2 shows the results on OK-VQA and A-OKVQA. With the accessible LLMs, we achieve the highest accuracy of 59.34% under single query setting, and enhance the accuracy of baselines by 2.15% on average. PromptCap achieves an accuracy of 60.45% on OK-VQA with `code-davinci-002` engine, which is currently inaccessible. While for the reproduction with `text-davinci-002`, we improve the performance of PromptCap by 0.9%. For A-OKVQA, we consistently observe improvements, and the average increment is1.25%.

Table 3 lists the results of previous methods on OK-VQA and A-OKVQA, where we report our ensemble results in accordance to Prophet. Within the realm of methods using LLMs, we further improve the best result by 0.2% for both dataset. Specifi-

| Methods | $n$-shot | Caption | Image Info. | Tag | Refine-$Q$ | Refine-$E$ | ACC |
|---|---|---|---|---|---|---|---|
| | 16 | PromptCap | Refined information | ✓ | ✓ | ✓ | 54.42 (a) |
| | 16 | PromptCap | Refined information | ✓ | ✓ | - | 52.25 (b) |
| Ours | 4 | PromptCap | Refined information | ✓ | ✓ | - | 50.19 (c) |
| | 4 | OFA-large | Refined information | ✓ | ✓ | - | 48.43 (d) |
| | 4 | OFA-large | Refined information | - | ✓ | - | 47.63 (e) |
| Ours w/o refinement | 4 | OFA-large | All the questions + BLIP-2 | - | - | - | 45.25 (f) |
| | 4 | OFA-large | Original questions + BLIP-2 | - | - | - | 45.17 (g) |
| Ours w/o BLIP-2 | 4 | OFA-large | Original questions | - | - | - | 44.86 (h) |
| BLIP-2 | 0 | - | - | - | - | - | 40.70 (i) |

Table 4: The ablation study of how different information affect the reasoning performance on OK-VQA. Ours: our proposed pipeline. Caption: the models used to generate the image captions. Image Info.: the resources of our image information, the "BLIP-2" in this column refers to the BLIP-2 answers, "Refined information" = applying refining on "All the questions + BLIP-2 answers". Tag: whether to use image tags. Refine-$Q$: applying refining to the "Image information $S$" in *Template-Query* in §3.3. Refine-$E$: applying refining to the "Image information $S^n$" in *Template-Few-shot* ($n$ in-context examples) in §3.3. Line (i) refers to directly using BLIP-2 for OK-VQA and the result is taken from Li et al. (2023). Please refer to §4.4 for detailed explanation and comparison.

| Scheme | Accuracy |
|---|---|
| Ensemble | 59.34 |
| Single-a | 58.53 -0.81 |
| Single-s | 58.48 -0.86 |
| Single-qa | 58.50 -0.84 |
| Single-q | 58.10 -1.24 |
| All | 58.03 -1.31 |
| (a) | |

| T | Accuracy | |
|---|---|---|
| | Ours | Prophet |
| T=1 | 59.3 | 57.9 |
| T=2 | 60.5 | - |
| T=3 | 61.1 | - |
| T=4 | 61.2 | - |
| T=5 | 61.3 | 61.1 |
| (b) | | |

Table 5: Ablation studies on OK-VQA. Results of our method in the two table refer to Prophet+Ours with 20-shot setting. (a): Comparison of refining schemes. Ensemble: 4-model ensemble. Single: -a/-s/-qa/-q refer to filter model trained using $a'$, $s'$, $q'a'$ and $q'$ respectively. All: using all information ($s + s'$) without refining. (b): Results of varying the number (T) of ensemble queries. T=1: no ensemble.

| Method | LLM | Accuracy |
|---|---|---|
| *Prophet* | | |
| Prophet + Ours | text-davinci-002 | 57.52 58.54 +1.02 |
| Prophet + Ours | LLaMA-13B | 50.76 53.08 +2.32 |
| Prophet + Ours | LLaMA-7B | 44.28 49.47 +5.19 |
| *PromptCap* | | |
| PromptCap + Ours | text-davinci-002 | 53.50 54.42 +0.92 |
| PromptCap + Ours | LLaMA-13B | 47.45 48.72 +1.27 |
| PromptCap + Ours | LLaMA-7B | 44.37 44.59 +0.22 |
| *PICa* | | |
| PICa + Ours | text-davinci-002 | 49.67 53.76 +4.09 |
| PICa + Ours | LLaMA-13B | 42.96 46.28 +3.32 |
| PICa + Ours | LLaMA-7B | 39.20 42.68 +3.48 |

Table 6: Results our method with different LLMs on OK-VQA. We use 16-shot for all methods in this table as a input of 16-shot is hitting the maximum input length of LLaMA.

cally, we achieve +1.59% on A-OKVQA compared to Prophet. For A-OKVQA, PromptCap achieves the previous highest accuracy, 59.6%. But we cannot directly apply our method upon it because its employed LLM engine is inaccessible now. For multimodal tuning setting, PaLM-E (Driess et al., 2023) and InstructBLIP (Dai et al., 2023) present the state of the art on OK-VQA and A-OKVQA, respectively. While PaLM-E is trained upon a 540B language model, PaLM (Chowdhery et al., 2022), it is over threefold of the largest LLM we use (175B). And InstructBLIP employs instruction tuning.

## 4.4 Ablation Study

We conduct the following ablation studies: 1) to verify that the gain of our method comes from properly integrating more image information; 2) to compare the effectiveness of different selection schemes in refinement module; 3) to investigate the impact of scaling up the number of generated

questions; 4) to analyse the impact of multi-query ensemble; and 5) to examine the consistency of our method across different LLMs.

**Appropriately integrated information helps to boost the performance.** We summarise the paradigm to solve VQA task into two categories. A VLM paradigm that directly uses a VLM (*e.g.*, BLIP-2 in Line (i) in Table 4), and an LLM-based paradigm where LLMs collaborate with VLMs to answer visual questions. Following the LLM-based paradigm, we progressively integrate new information and apply refining methods from Line (h) to (a) in Table 4, and observe continual enhancements.

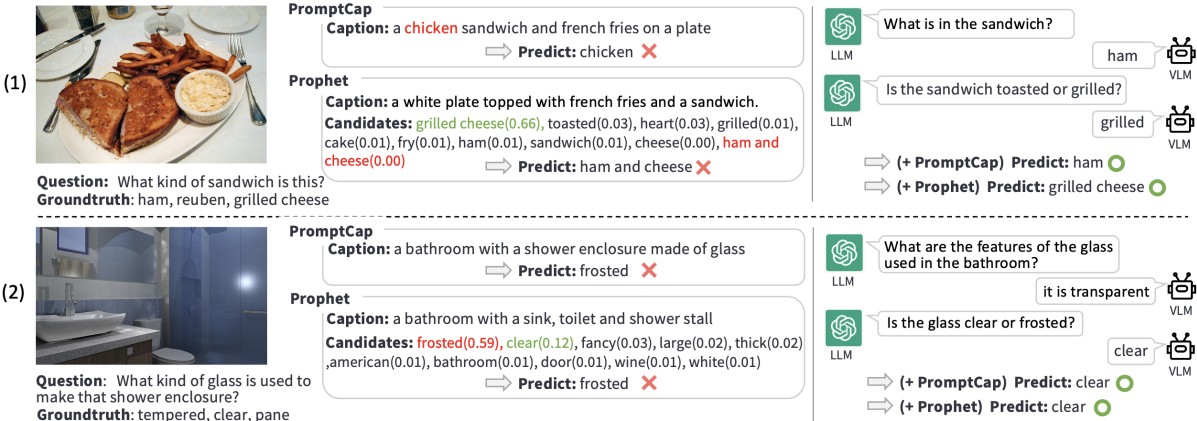

Figure 3: Cases compared to Prophet and PromptCap without applying our method. The frames titled by "Prompt-Cap"/"Prophet" depict the results given by these two baselines in our reproduced version. The information leading to incorrect answers are marked in red.

Line (h) represents results for 4-shot setting following the PICa template. While Line (g) implements our template, *Template-Few-shot* with $n = 4$. The BLIP-2 answers are included in "Image information" in *Template-Few-shot* and *Template-Query*. The participation of BLIP-2 raises the accuracy by 0.31%. When we feed all generated information (Line (f)), the accuracy only marginally improves by 0.07%. Notably, applying the refinement module to the query instance (Line (e)) results in a significant 2.38% accuracy boost. Line (d) introduces image tags and contributes an additional 0.8% improvement compared to Line (e). Extending to 16-shot setting further boosts the result by 2.06%. While the previous lines only apply information refining to the query instance, we go further in Line (a) by refining the in-context examples. This contributes to an increment of 2.17% compared to Line (b), and is more influential than adding tags, increasing the number of shots and replacing the caption type.

In conclusive, the performance benefits from adding more image information. Our method to seek and refine image information makes a substantial contribution in this regard.

**Effectiveness of filtering schemes.** We evaluate the performance of different selection schemes in the refinement module, including (1) Ensemble: selecting according to ensemble filters as proposed in §3.2, (2) Single: selecting according to single filter, including filters trained using answers, question, summaries, and question-answer pairs (qa), and (3) All: selecting all generated questions $q'$ and the original question $q$ without filtering.

As shown in Table 5 (a), filters help improve the performance against directly selecting all the questions without filtering. However, single filters make only minor contributions to the final accuracy because each of them addresses only a single aspect of the criteria.

**Scaling up the number of generated questions.** Intuitively, scaling the amount of generated questions will contribute to extra gain. To verify this, we doubled the number of generated questions to 6. By applying the same selection strategy under 20-shot single query setting, we obtain a final accuracy of 59.68% on OK-VQA, which is slightly higher (+0.34%) than generating only 3 questions.

**Impact of multi-query ensemble.** PICa and Prophet exhibit improvements of multi-query ensemble by prompting LLM for 5 times with different in-context examples. We investigate the influence of multi-query ensemble on our method. As shown in Table 5 (b), although the accuracy of our method increases along with the number of ensemble queries, the gap between ours and Prophet's are narrowed. As the in-context examples are arranged according to the relevance to test instance, the more examples we use, the less coherent to the test instance they will be. Thereby, noises could be introduced with the progressive increase of ensemble queries. Similarly, Chen et al. (2023) also observe a decline in performance when continuously increase the number of ensemble queries.

**Consistency of our method across LLMs.** Different LLMs largely affect the results (Shao et al., 2023; Hu et al., 2023a). We investi-

gate if our method is generalizable across LLMs trained in different ways and with different scales, including LLaMA-7B, LLaMA-13B, and `text-davinci-002` (175B). Table 6 proves the effectiveness of our method on both InstructGPT (`text-davinci-002` engine) and LLaMA. Results also demonstrate the robustness of our method on the scales of LLM, ranging from 7B to 175B. We notice that our method introduce less improvement on PromptCap compared to Prophet and PICa. We suppose that the captions of PromptCap are derived from the question, and they seem to be more determinate as shown in Figure 3, which easily dominates the reasoning and impair the attendance of other information.

## 5 Case Study

We conduct comprehensive analysis of the cases, and observe that our method exhibits the following capabilities: 1) to unveil a better level of detail as stated in the §3.1; 2) to rectify inaccurate caption information; and 3) to minimize the ambiguity of provided information, *e.g.*, captions and candidates. Cases are demonstrated in Figure 3. We compared the results of baselines, PtomptCap and Prophet, with and without applying our method.

**Unveiling Missing Details.** Our generated questions bridge the information gap between the question and the image caption by revealing missing details necessary to answer the question. An shown in (2) of Figure 3, the question asks about the "kind of glass", but relevant features are not included in the PromptCap caption. The absence of detail leads to an improper prediction, "frosted". However, the two questions in our method pinpoint the detailed features, "transparent" and "clear", and contribute to a correct prediction. These imply the effectiveness of our generated questions.

**Rectifying Inaccurate Information.** Our method can rectify the misleading information provided in the captions. In the first case shown in Figure 3, we correct the wrong message given by PromptCap that it is not a "chicken" sandwich by the question "what is in the sandwich" with answer "ham".

**Minimizing the Ambiguity.** By engaging more image information, our method provides evidence to support the correct candidates in Prophet and the proper captions of PromptCap, thereby enhancing the confidence and the reliability of accurately provided information in these baselines. In Figure 3, the candidates of Prophet in (1) properly

fit the image. However, the LLM does not follow the given confidence and selects the least confident one. In contrast, Figure 3 (2) demonstrates a situation where the most confident candidate is not the correct answer. In this two scenarios, our method supports the correct answer with more detailed information.

## 6 Conclusion

In this paper, we focus on open-domain knowledge-based VQA and propose a model agnostic framework that successfully unveils missing detail during the image-to-text transformation. Our method acquires the ability to rectify inaccurate information generated by captioning models, and the ability to minimize the ambiguity of the converted textual information for further improvements. Our method can be applied upon existing baselines, and achieves average gains of 2.15% on OK-VQA and 1.25% on A-OKVQA over the baselines. Ablation studies show that our method attain consistent improvements across different LLMs.

## Limitations

In this work, we demonstrate the effectiveness of our framework on OK-VQA and A-OKVQA, and show a consistent improvement across different LLMs. However, we do not verify the feasibility of our idea on other vision-language tasks that also require knowledge, such as visual commonsense reasoning. Intuitively, the paradigm to prompt LLMs to uncover missing image details can be applied to a wild range of VL tasks. While the questions in our framework are generated independently, further challenges include to progressively ask informative question upon the previous questions and acquire the accurate answers. We hope that our work will encourage further investigations that explore the capabilities of LLMs in the VL realm.

## Acknowledgement

This work is supported by the National Key R&D Program of China (2022ZD0160502) and the National Natural Science Foundation of China (No. 61925601, 62276152). We thank Siyu Wang for her participation in this work, and appreciate all the reviewers for their insightful suggestions.

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

## A Details for Gathering Training Data for Refinement Module

The process for constructing the training data for refinement module as described in §3.2. We denote our generated image information as a set $\mathcal{I}$, containing the four types of generated image information, $q'$, $a'$, $q'a'$ and $s'$. $\boldsymbol{Q}$ is the set of all the original visual questions, and $\boldsymbol{I}$ refers the corresponding images. $\boldsymbol{y}_z$ is resulting training labels used in §3.2, where $y_{z^k}$ refers to the label for a single generated information. The soft accuracy score $\text{Acc}_{soft}(a)$ is computed following Goyal et al. (2017).

---

**Algorithm 1** Pipeline for Supervision Gathering
**Input:** Image, Question and Generated Image Information $\{\boldsymbol{I}, \boldsymbol{Q}, \mathcal{I}\}$
**Output:** Image Information ($\mathcal{I}$), Original Questions ($\boldsymbol{Q}$) and Images ($\boldsymbol{I}$) with labels ($\boldsymbol{y}_z$)
**Require:** $\mathcal{P}_{i,q}$ is the prompt for answer reasoning regarding question $q$ and image $i$.

1: **procedure** GET LABELS($\boldsymbol{y}_z$)
2:     **for** $q \in Q$ and $i \in I$ and $\mathcal{I} \in \boldsymbol{\mathcal{I}}$ **do**
3:         $a \leftarrow \text{LLM}_{reason}(\mathcal{P}_{i,q}(\mathcal{I}))$
4:         $acc_q \leftarrow \text{Acc}_{soft}(a)$
5:         **if** $acc_q > 0$ **then**
6:             $y_z \leftarrow 1$
7:         **else**
8:             $y_z \leftarrow 0$

---

## B The Declaration of Trainable Modules in Our Pipeline

These models are frozen in our entire pipeline: the captioning model $\mathcal{M}_c$, the VQA model $\mathcal{M}_a$ and the LLMs (for question generation, summarization and reasoning).

The refinement module in Figure 2 requires training, which includes 4 parts:

- An image encoder (for image): to encode the original VQA image;

- A question encoder (for text): to encodes the original VQA question;

- An information encoder (for text): to encode our generated image information (denoted as "Info encoder" in Figure 1);

- A filter: an MLP-based network that evaluates the helpfulness of the obtained image information towards answering the visual questions.

## C Full List of Results

We provide results of previous methods on OK-VQA and A-OKVQA in Table 7 and Table 8. The current state-of-the-art on OK-VQA is 66.1 and is contributed by a 562B pre-trained multimodal model, including a 540B language model and a 22B vision encoder. We achieve the highest score in both methods querying external KBs and methods with LLMs. InstructBLIP (Dai et al., 2023) achieves the SOTA of A-OKVQA with multimodal instruction tuning.

| Method | Accuracy |
|---|---|
| *Multimodal pre-train* | |
| LAMOC11B (Du et al., 2023) | 40.3 |
| BLIP-2 (FlanT5-XL) (Li et al., 2023) | 40.7 |
| BLIP-2 (FlanT5-XXL) (Li et al., 2023) | 45.9 |
| OFA-large (Wang et al., 2022) | 49.4 |
| Unified-IO (2.8B) (Pai et al., 2000) | 54.0 |
| Flamingo (80B) (Alayrac et al., 2022) | 57.8 |
| InstructBLIP (Dai et al., 2023) | 62.1 |
| PaLM-E (562B) (Driess et al., 2023) | 66.1 |
| *Methods querying external KBs* | |
| ConceptBERT (Gardères et al., 2020) | 33.7* |
| KRISP (Marino et al., 2021) | 38.9 |
| Vis-DPR (Luo et al., 2021) | 39.2 |
| MAVEx (Wu et al., 2022b) | 40.3 |
| VLC-Bert (Ravi et al., 2023) | 43.1 |
| TRiG (Gao et al., 2022) | 50.5 |
| RA-VQA (Lin and Byrne, 2022) | 54.5 |
| REVEAL (Hu et al., 2022) | 59.1 |
| *Methods with LLMs* | |
| Img2Prompt175B (Guo et al., 2023) | 45.6 |
| KAT (ensemble) (Gui et al., 2022) | 54.4 |
| KAT-full EnFoRe (ensemble) (Wu and Mooney, 2022) | 55.2 |
| PICa-Full (Yang et al., 2022) | 48.0 |
| REVIVE (Lin et al., 2022) | 58.0 |
| PromptCap$^{\diamond}$ (Hu et al., 2023a) | 60.4 |
| Prophet (Shao et al., 2023) | 57.9 |
| Prophet+ours | 59.3 (+1.4) |
| Prophe (ensemble) (Shao et al., 2023) | 61.1 |
| Prophet+ours (ensemble) | **61.3** (+0.2) |

Table 7: Comparisons to other methods on OK-VQA. * indicates results reported on OK-VQA v1.0. $^{\diamond}$ refers to method with currently deprecated LLMs. InstructBLIP refers to InstructBLIP(Vicuna-7B). The highest accuracy within methods using LLMs is **bolded** and the highest accuracy within the other two scopes are underlines.

| | Method | Accuracy |
|---|---|---|
| *Fine-tune* | Pythia (Schwenk et al., 2022) | 25.2 |
| | ViLBERT (Schwenk et al., 2022) | 30.6 |
| | LXMERT (Schwenk et al., 2022) | 30.7 |
| | ClipCap (Schwenk et al., 2022) | 30.9 |
| | KRISP (Marino et al., 2021) | 33.7 |
| | LAMOC (Du et al., 2023) | 37.9 |
| | VLC-BERT (Ravi et al., 2023) | 45.0 |
| | GPV-2 (Schwenk et al., 2022) | 48.6 |
| | Unified-IO (Pai et al., 2000) | - |
| | REVEAL (Hu et al., 2022) | 52.2 |
| | InstructBLIP (Dai et al., 2023) | **64.0** |
| *In-context* | Img2Prompt175B (Guo et al., 2023) | 42.9 |
| | assistGPT (Gao et al., 2023) | 44.3 |
| | PromptCap$^\diamond$ (Hu et al., 2023a) | 56.3 |
| | Prophet (Shao et al., 2023) | 58.2 |
| | Prophet+ours | 59.8 |

Table 8: Comparisons to other methods on A-OKVQA DA task. $\diamond$ refers to method with currently deprecated LLMs. InstructBLIP refers to InstructBLIP(Vicuna-7B). The highest accuracy is **bolded** and the second best is underlines.

## D Ensemble with different numbers of shots

The results of ensemble queries are listed in Table 9. We notice that our method improved the result of Prophet by 1.43% in 20-shot single query setting. But the increment decreases to 0.2% when applying ensemble. Also, similar pattern can be found in 16-shot setting. The increments from T=1 to T=3 are relatively significant, which are 1.8% (20-shot) and 2.1% (16-shot). While continuously increasing the number of ensemble queries from T=3 to T=5, the increments decrease to 0.2% and 0.1%. Because the in-context examples are arranged according to the relevance to the test instances, these phenomenon could result from engaging irrelevance examples when enlarging the number of queries.

| T | 20-shot | | 16-shot | |
|---|---|---|---|---|
| | Ours | Prophet | Ours | Prophet |
| T=1 | 59.3 | 57.9 | 58.5 | 57.5 |
| T=2 | 60.5 | - | 60.1 | - |
| T=3 | 61.1 | - | 60.6 | - |
| T=4 | 61.2 | - | 60.8 | - |
| T=5 | 61.3 | 61.1 | 60.9 | 60.8 |

Table 9: 20-shot and 16-shot results of the number of ensemble queries (T) on OK-VQA. T=1 means no ensemble.

We conduct further examination on the ensemble behavior on OK-VQA dataset and find out that Prophet indeed benefits more from ensemble compared to the other baselines. Table 10 shows the results of the single query setting and the ensemble setting. For Prophet, the gain of the ensemble setting over the single setting is 3.19% (Line 5), which is more than doubled compared to results on Line 1 to Line 4, and is nearly doubled compared to Line 6. This shows an inconsistency regarding the gain of ensemble setting over single setting, and supports our hypothesis that Prophet benefits more from the ensemble setting compared to other methods.

| Methods | Single | Ensemble | Gain |
|---|---|---|---|
| PICa | 49.67 | 51.36 | 1.69 |
| + Ours | 53.76 (+4.09) | 54.91 (+3.55) | 1.15 |
| PromptCap | 53.50 | 54.16 | 0.66 |
| + Ours | 54.42 (+0.92) | 55.01 (+0.85) | 0.59 |
| Prophet | 57.91 | 61.10 | **3.19** |
| + Ours | 59.34 (+1.43) | 61.30 (+0.20) | 1.96 |

Table 10: Single and Ensemble performances of our methods and baselines. Gain refers to the increment of ensemble result over single result.

## E Experiments with Dense Captions

In our framework, the LLM perceives the image from two parts of the prompt: "Caption" and "Image Information" (as the templates described in §3.3). We incorporate the dense captions (denoted as GRiT (Wu et al., 2022a)) and conduct two types of experiments with the PICa pipeline:

- Dense captions directly as "Caption";

- Dense captions as a type of "Image Information".

As shown in Table 11, the first two lines show the influence of dense captions and general captions. Simply replacing the OSCAR captions by GRiT captions reduces the accuracy from 49.67% to 46.16%. Adding GRiT to PICa as the "Image Information" (line 4) also impairs the original PICa performance (line 1) by 0.24%.

Lines 3, 4 and 5 present the performances of different types of "Image Information". Notably, replacing our information by GRiT's dense captions decreases the performance by 4.33% (comparing line 4 to line 3). While combining Ours and GRiT captions as the "Image Information" (line 5) reduces the accuracy by 1.21% compared to using only Ours "Image Information" (line 3).

To conclude, dense captions introduce noise and degrade the accuracy, regardless of the role as either "Caption" or "Image Information". In contrast,

utilizing the image information obtained and refined by our method consistently yields the best results.

| Methods | Caption | Image Info. | ACC |
|---|---|---|---|
| PICa | OSCAR | - | 49.67 |
| PICa | GRiT | - | 46.16 (-3.15) |
| +Ours | OSCAR | Ours | 53.76 |
| +D.C. | OSCAR | GRiT | 49.43 (-4.33) |
| +D.C.+Ours | OSCAR | GRiT+Ours | 52.55 (-1.21) |

Table 11: Experimental results with dense captions (GRiT). "Caption" refers to the type of caption used in *Template-Few-shot* and *Template-Query* in §3.3. "Image Info." refers to the type of image information used in *Template-Few-shot* and *Template-Query* in §3.3.

## F   Prompt Templates

In this section, we provide detailed prompt template for question generation, summarization, and in-context learning for visual question reasoning. Since the input length of LLMs are limited, to present LLMs with more potential image information and relevant knowledge, we remove the separators ("===") used in PICa and its followers.

### F.1   Prompt Templates for Question Generation

Here is the prompt template and an example for question generation described in §3.1. The number of questions to generate is 3, and the test instance is marked in blue. The template and example are as follows:

---

**Prompt Template for Question Generation**

```
Please decompose the TARGET-QUESTION into
3 questions that can be answered via
commonsense knowledge.  The sub-questions
should not mention another sub-questions.
You can use information from the CAPTION.\n
TARGET-QUESTION: q^n\n
Caption: C^n\n
Sub questions: 1. q'^n_1. 2. q'^n_2, ...\n
TARGET-QUESTION: q\n
Sub questions:\n
```

---

**An Example for Question Generation**

```
Please decompose the TARGET-QUESTION into
3 questions that can be answered via
commonsense knowledge.  The sub-questions
should not mention another sub-questions.
You can use information from the CAPTION.\n

TARGET-QUESTION: What is the hairstyle of
the blond called?\n
Caption: Two women tennis players on a
tennis court.\n
Sub questions: 1. It this hairstyle long
or short? 2. What are the notable features
of the hairstyle? 3. What hairstyle are
common for women player when they are
playing tennis\n

TARGET-QUESTION: How old do you have
to be in canada to do this?\n
Caption: a couple of people are holding up
drinks.\n
Sub questions: 1. Why are people holding
up drinks? 2. What is the restriction of
age to drink in Canada? 3. What are people
drinking?\n

TARGET-QUESTION: When was this piece
of sporting equipment invented?\n
Caption: A man in a wetsuit carrying a
surfboard to the water.\n
Sub questions: 1. What is the man carrying
with him?  2.  What is the purpose of
the sporting equipment?  3.  What is the
history of the invention of the sporting
equipment?\n

TARGET-QUESTION: What hair style does
the child have?\n
Caption: a little girl with short hair
talking on a cell phone.\n
Sub questions:\n
```

---

### F.2   Prompt Templates for Summarization

In the refinement module, we summarize the generated questions with corresponding answer into narrative expressions for further process. Here is the prompt template and an example for information summarization described in §3.2, the test instance is marked in blue:

---

**Prompt Template for Summarization**

```
Please summarise the following question
and corresponding answer into a description
sentence.\n
Q: q^n\n A: a^n\n Summary: 1. q'^n_1. 2. q'^n_2,
...\n
Q: q\n A: a^n\n Summary:\n
```

### F.3 Prompt Templates for Reasoning

We employ few-shot in-context learning for answer reasoning. Here is the prompt template described in §3.3, the test instance is marked in blue:

We implement our method with different baselines according to the their default settings for image representation. PICa employs captions with tags as image representation; PromptCap uses thire question-aware captions; and Prophet provides extra answer candidates. There are the examples for PICa+ours, PromptCap+ours and Prophet+ours, our refined information is bolded in the template, and the test instance is marked in blue:

## An Example for Reasoning with PromptCap

Answer the questions using the provided image information, captions and extra commonsense knowledge. Answers should be no longer than 3 words:\n

**Image information:** the person is skiing; the person is wearing skis on their feet; cross country skiing is a popular activity while skiing.\n
Caption: A person skiing on a snowy road.\n
Question: What is this person doing?\n
Answer: cross country ski

**Image information:** the person is wearing skis; cross country skis are one of the equipment options for this activity; Snow conditions impact travel safety during this activity.\n
Caption: A person skiing down a snowy hill.\n
Question: What is this person doing?\n
Answer: ski

…

**Image information:**The women steps over the water; Water freezes when it gets cold; The area will change when the temperature reaches 0 degrees.\n
Caption: a woman on skis in the snow\n
Question: If it gets cold enough what will happen to the area being stepped over?\n
Answer:

## An Example for Reasoning with Prophet

Answer the questions using the provided image information, captions, candidate answers and extra commonsense knowledge. Each candidate answer is associated with a confidence score within a bracket. The true answer may not be included in the candidate answers. Answers should be no longer than 3 words:\n

**Image information:** the person is skiing; the person is wearing skis on their feet; cross country skiing is a popular activity while skiing.\n
Caption: A man is cross country skiing through a forrest in winter.\n
Question: What is this person doing?\n
Candidatew: ski (0.98), cross country ski (0.63), skiis (0.13), hike (0.11), snow (0.09), cross country (0.02), skiing (0.01), snowboard (0.00), camp (0.00), cold weather (0.00)\n
Answer: cross country ski

**Image information:** the person is wearing skis; cross country skis are one of the equipment options for this activity; Snow conditions impact travel safety during this activity.\n
Caption: A man on skis riding through the snow. \n
Question: What is this person doing?\n
Candidatew: ski (0.99), snow (0.66), sky (0.15), water (0.03), skiis (0.02), ski pole (0.01), downhill (0.01), snowboard (0.00), hill (0.00), commuter (0.00)\n
Answer: ski

…

**Image information:**The women steps over the water; Water freezes when it gets cold; The area will change when the temperature reaches 0 degrees.\n
Caption: A woman on skis in the snow near a tree.\n
Question: If it gets cold enough what will happen to the area being stepped over?\n
Candidatew: fall (0.04), crash (0.02), break (0.01), avalanche (0.01), death (0.01), cold (0.00), freeze (0.00), autumn (0.00), oxygen (0.00), drown (0.00)\n
Answer: