# OpenReview forum: "Filling the Image Information Gap for VQA: Prompting Large Language Models to Proactively Ask Questions"
_EMNLP/2023/Conference — EMNLP 2023 Findings_

### Official Review · Reviewer_cxzi · 2023-07-31

**Typos Grammar Style And Presentation Improvements:** Appendix A, line 851, should be y_{z^…
**Soundness:** 3

**Excitement:**

2: Mediocre: This paper makes marginal contributions (vs non-contemporaneous work), so I would rather not see it in the conference.

**Missing References:**

There are certain works that directly use caption as additional information and use the same VLM (instead of LLM) to answer the vqa tasks should be metioned somehow.

@misc{awal2023investigating,
      title={Investigating Prompting Techniques for Zero- and Few-Shot Visual Question Answering},
      author={Rabiul Awal and Le Zhang and Aishwarya Agrawal},
      year={2023},
      eprint={2306.09996},
      archivePrefix={arXiv},
      primaryClass={cs.CV}
}

@misc{guo2023images,
      title={From Images to Textual Prompts: Zero-shot VQA with Frozen Large Language Models},
      author={Jiaxian Guo and Junnan Li and Dongxu Li and Anthony Meng Huat Tiong and Boyang Li and Dacheng Tao and Steven C. H. Hoi},
      year={2023},
      eprint={2212.10846},
      archivePrefix={arXiv},
      primaryClass={cs.CV}
}

**Paper Topic And Main Contributions:**

This paper concentrates on knowledge-based Visual Question Answering tasks. It enhances existing methods by introducing a novel framework that extracts intricate, question-related caption from images. The core strategy involves deconstructing questions into subquestions based on the given captions and query, extracting answers for each subquestion, and thereafter generating a comprehensive caption that incorporates all this information. This enhanced information, coupled with the original question and caption, is fed into the Language Learning Model (LLM). The infusion of this supplemental image information amplifies the LLM's capacity for reasoning, enabling it to generate more precise and informed answers.

The primary contributions are a unique, comprehensive framework and a experimental results that provide extensive validation of the methodology proposed.

**Questions For The Authors:**

How do you judge the improvement compared with complexity of the method?

Please answer my concers in `Reasons To Reject` section, if most of question are resovled, the score could be increased.

**Reasons To Accept:**

This paper convincingly illustrates the necessity of providing nuanced, query-specific information to complement the coarse, generalized image captions diretly output from caption model without any query information. It recognizes a shortcoming in existing approaches that convert image to text for application in large language models to solve visual question-answering tasks. The one-off conversion process often loses important details, underscoring the importance of query-related information to infuse sufficient visual details into the Language Learning Model .

The proposed framework is thoughtfully and intuitively designed, reflecting an in-depth understanding of the problem at hand.

The research is supported by thorough and comprehensive experiments. It includes a comparative analysis with most related works, and an exhaustive ablation study, all of which affirm the effectiveness of the proposed method.

**Reasons To Reject:**

The study presents a commendable motivation and idea, yet the implementation of the method seems disproportionately complex given the relatively marginal improvement. The method encompasses several modules, including a caption model, VQA model, filter, and large language models. However, the advancement over 'Prophet' does not appear to warrant the complexity and increased inference time of the method.

The articulation of the method and experiments is not clear. For instance, (1) at line 214, the method for calculating scores in Table 1 is unclear; (2) lines 264-265 lack sufficient detail, relegating crucial information to the appendix and leaving the motivation and methods for obtaining ground-truth VQA accuracy unclear; in Appendix A, P_{i,q} and ACC_{soft} aren't sufficiently explained; (3) at lines 275-276, it is unclear how z_q is calculated. While Equation 8 appears to use all information, it remains unexplained if only 'q' is used to get z_q; (4) in Section 3.3, the example prompt is not consistent with Figure 2, where the figure does not display `caption`; (5) the 'Prophet' results in Tables 2 and 3 are inconsistent; (6) for Table 4, the application of answers from 'Blip2' remains unclear. Are they used for few-shot examples? If so, how does 'ourts-w/o' function? (7) Table 4 should provide an explanation for the meaning of 'n' and 'E' in its caption; (8) the statement at lines 414-415 claiming an improvement from 40.7 to 47.63 seems unreasonable, as 40.7 directly uses 'Blip2', a VLM, while 47.63 uses a different method involving LLM. The improvement could be attributed not just to more detailed information but also enhanced reasoning ability.

In the case study, Figure 3 (1) suggests that 'Prophet's' output of 'grilled cheese' with a probability of 0.66 should indeed be the correct answer

**Reproducibility:**

3: Could reproduce the results with some difficulty. The settings of parameters are underspecified or subjectively determined; the training/evaluation data are not widely available.

**Reviewer Confidence:**

4: Quite sure. I tried to check the important points carefully. It's unlikely, though conceivable, that I missed something that should affect my ratings.

---

> ### Author Rebuttal · Authors · 2023-08-29
>
> **Questions 1: Complexity vs performance**
>
> **Response 1:**
>
> - The complexity:
>     - Prophet also needs a frozen captioning model and requires to train different VQA models for different benchmark.
>     - In our framework, we train the refinement module in Figure 2, but freeze the VQA and captioning models in the inquery module in Figure 2. We conduct ablation study to show the effectiveness and necessity of our refinement module. As reported in Table 2 and Table 5(a) in our paper, using all the information **without refining** ("All" in Table 5(a)) results in an accuracy of 58.03% (only +0.12% compared to Prophet). By introducing our refinement module, the accuracy increases to 59.34% ("Prophet+Ours" in Table 2, +1.43% compared to Prophet).
>
> - The inference speed:
>
>     The inference speeds of Prophet and our method are nearly identical. When requesting a OpenAI API (for GPT-3.5), both Prophet and our method cost 1.1-1.2 second/instance in the 20-shot, single query setting. In the ensemble setting, both speeds increase to 4.5-5.2 second/instance for ensemble setting.
>
>     Excluding the influence of API, both the VQA model in Prophet and the refinement module in our method require approximately 0.02 second/instance.
>
> - The improvement of our method:
>
>     We would like to further emphasize the improvements of our method compared to Prophet in Table 3 in our paper as follows:
>
>     - For A-OKVQA, our single query result (57.9%, "Prophet+Ours" in Table 3) is already comparable to the ensemble result of Prophet (58.2%, "Prophet (ensemble)") with a significantly lower inference cost. Furthermore, our ensemble performance reaches 59.8% ("Prophet+Ours (ensemble)"), marking a 1.6% improvement.
>
>     - For OK-VQA, we obtain an increment of 1.4% over Prophet in the single query setting ("Prophet+Ours"). In the ensemble setting, we can observe a performance improvement of 0.2% ("Prophet+Ours (ensemble)").
>         - One factor that might contribute to this reduced improvement in the ensemble setting is the inherent characteristics of the Prophet method. Prophet relies on a fine-tuned VQA model to generate answer candidates from a closed-set of predefined answers (all the answers present in the training dataset).
>         - This constraint is particularly noteworthy for OK-VQA, given its lack of answer diversity compared to A-OKVQA. The OK-VQA dataset has 11,508 distinct answer categories, with each question having a maximum of 5 different answers. On the other hand, the A-OKVQA dataset contains 29,631 unique answer categories, with each question having up to 10 distinct answers.
>         - By integrating more few-shot examples, the answer candidates will present a dataset bias. For OK-VQA dataset, we hypothesize that Prophet can benefit from the biased candidates in the ensemble setting.
>
>     - We conduct further examination on the ensemble behavior on OK-VQA dataset and find out that Prophet indeed benefits more from ensemble compared to the other baselines. The table below shows the results of the single query setting and the ensemble setting. For Prophet, the gain of the ensemble setting over the single setting is 3.19% (Line 5), which is more than doubled compared to results on Line 1 to Line 4, and is nearly doubled compared to Line 6. This shows an inconsistency regarding the gain of ensemble setting over single setting, and supports our hypothesis that Prophet benefits more from the ensemble setting compared to other methods.
>
>         |No.|    Method     | Single        | Ensemble      | Gain of Ensemble over Single|
>         |:--|:--------------|:--------------|:--------------|:-----------------:|
>         | 1 | PICa          | 49.67         | 51.36         | 1.69             |
>         | 2 | PICa+ours     | 53.76 (+4.09) | 54.91(+3.55)  | 1.15             |
>         | 3 | PromptCap     | 53.50         | 54.16         | 0.66             |
>         | 4 | PromptCap+ours| 54.42 (+0.92) | 55.01 (+0.85) | 0.59             |
>         | 5 | Prophet       | 57.91         | 61.10         | **3.19**         |
>         | 6 | Prophet+ours  | 59.34 (+1.43) | 61.30 (+0.20) | 1.96             |
>
>     - Therefore, we consider the reduced improvement in the OK-VQA + ensemble setting to be a special case, whereas in most other cases our approach leads to significant improvements compared to Prophet.
>
> **Question 2: Questions about the method and experiments**
>
> **Response 2:**
>
> - **(1) Line 214, the method for calculating scores in Table 1 in our paper.**
>
>     The reported scores are the soft accuracy of VQA according to the paper [VQA: Visual Question Answering](https://arxiv.org/pdf/1505.00468.pdf), and are implemented according to [github link](https://github.com/GT-Vision-Lab/VQA). The scores in Table 1 in our paper are calculated under the 4-shot setting. Each column represents:
>
>     - "Original": reasoning with the original question only;
>     - "All": using all the information for reasoning, including the summarization of the original question and its BLIP-2 answer and the summarizations of all the generated question-answer pairs for an instance;
>     - "Random": randomly select one summarization from the "All" setting above;
>     - "Best": the upper bound score among information in the "All" setting. We use the highest accuracy of each instanse as the "Best" score.
>
>    This table demonstrates the effectiveness of the generated questions together with BLIP-2 answers, and the necessity of a refinement module.
>
> - **(2) Lines 264-265 and details in Appendix A in our paper.**
>
>     Please refer to Appendix D.3 for examples of the prompt $P_{i,q}$. The $ACC_{soft}$ for each intance is calculated as:
>
>      $ Acc_{soft}(ans) = min(\{ \frac{number \quad of \quad matching \quad answers}{3}, 1\}) $
>
>     We will move the relevant content back to Section 3 in our paper for better understanding if permitted extra space.
>
> - **(3) Lines 275-276, how is $z_q$ calculated?**
>
>     $z_q$ is the contrubition score of the original question *q* with BLIP-2 answers. It is not calculated with *q* only. We use the summarization of the original question *q* together with its BLIP-2 answer as the textual input for the filter.
>
> - **(4) The example prompt is not consistent with Figure 2, where the figure does not display caption.**
>
>     Sorry for causing the confusion. We omit the caption in Figure 2 because it is a common field among our method and the baselines. We will put the caption back in the answering module in Figure 2 in our revised version. Thanks for pointing that out.
>
> - **(5) The inconsistent results in Tables 2 and 3 in our paper.**
>
>     Most of the results in Table 2 in our paper are reported with the single query setting, except for those marked by '\*' because we cannot find any single query setting results publicly. Since most of the methods report their single query results, we also report the single query results in Table 2 in our paper for fair comparison. We stated in the caption of Table 2 in our paper that '\*' refers to multi-query ensemble results. In Table 3 in our paper, the single query results of Prophet (57.9%) and Prophet+ours (59.3%) are the same as those in Table 2.
>
> - **(6) For Table 4, the application of answers from 'Blip2' remains unclear.**
>
>     Sorry for the unclear design of the table. We reform Table 4 as below:
>
>     | Method | Number of Shots| Caption | Image Information | Tags | Refinement on Query Instance | Refinement on Few-shot Examples | ACC |
>     |---------------------|---|-----------|---------------------------|-----|-----|-----|----------|
>     | Ours                | 16| PromptCap | Refined information    |&#10004;|&#10004;|&#10004;|  54.42(a)|
>     | Ours                | 16| PromptCap | Refined information       |&#10004;|&#10004;| -   |  52.25(b)|
>     | Ours                | 4 | PromptCap | Refined information       |&#10004;|&#10004;| -   |  50.19(c)|
>     | Ours                | 4 | OFA-Large | Refined information       |&#10004;|&#10004;| -   |  48.43(d)|
>     | Ours                | 4 | OFA-Large | Refined information       | -   |&#10004;| -   |  47.63(e)|
>     | Ours w/o refinement | 4 | OFA-Large | All the questions + BLIP-2 answers| -   | -   | -   |  45.25(f)|
>     | Ours w/o refinement | 4 | OFA-Large | Original question + BLIP-2 answers| -   | -   | -   |  45.17(g)|
>     | Ours w/o BLIP-2     | 4 | OFA-Large | Original question         | -   | -   | -   |  44.86(h)|
>     | BLIP-2              | 0 | -         | -                         | -   | -   | -   |  40.70(i)|
>
>     Recalling the prompt template for answer reasoning in Section 3.3 in our paper, the template consists of two parts:
>
>     - Few-shot in-context examples (Line 287 to Line 288 in our paper), denoting as ***Template-Few-shot***:
>
>             Image information: S^n \n Caption: C^n \n Question: q^n \n Answer: a^n
>
>     - A query instance (Line 290 to Line 291 in our paper), denoting as ***Template-Query***:
>
>             Image information: S \n Caption: C \n Question: q \n Answer:
>
>     With the in-context examples, the LLM is required to predict the answer for the question q in the query instance.
>
>     Here are some explanations of the table above:
>
>     - "Caption": the models used to generate the image captions;
>     - "Image Information": the resources of our image information, "Refined information" = applying refining on "All the questions + BLIP-2 answers";
>     - "Tags": whether to use image tags. According to [PICa](https://arxiv.org/pdf/2109.05014.pdf), image tags are place right after the captions in the template above;
>     - "Refinement on Query Instance": applying refining to the "Image information S" in ***Template-Query*** mentioned above;
>     - "Refinement on Few-shot Examples": applying refining to the "Image information S^n" in ***Template-Few-shot*** mentioned above;
>     - Line (i): BLIP-2 refers to directly using BLIP-2 for OK-VQA.
>     - **Response to the questions of the application of BLIP-2 answers: Are they used for few-shot examples? If so, how does 'ours-w/o' function?**
>         - The BLIP-2 answers are not used for few-shot examples. They are used for generating the image information ("S^n" and "S" in the template above) used in the "Image information" field.
>         - The "ours-w/o" in the original Table 4 in our paper is moved to Line (h) as "Ours w/o BLIP-2" in the nre table above. The image information ("S^n" and "S" in the template above) refer to the summarization from quesitons and the corresponding BLIP-2 answers. While in Line (h), we directly place the original questions without BLIP-2 answers in the "Image information" field in both ***Template-Query*** and ***Template-Few-shot***.
>
>     Here are the brief conclusions of this table:
>
>     - Line (e) vs Line (f): refinement module significantly improves the accuracy (+2.38%);
>     - Line (g) vs Line (h): BLIP-2 answers help (a bit) the LLM to reason for answers (+0.31%);
>     - Line (d) vs Line (e): tags are helpful (+0.80%);
>     - Line (b) vs Line (c): increasing the number of shots significantly improves the accuracy (+2.06%);
>     - Line (a) vs Line (b): applying refinement on the few-shot examples markedly boost the accuracy (+2.17%).
>
>     This is the main results of the ablation study that claims the importance and necessity of every details in our framework.
>
> - **(7) Lines 414-415, an improvement from 40.7 to 47.63 seems unreasonable.**
>
>     Sorry for causing the confusion. As we stated in our paper in Line 414-416, "there is a continuous gain from 40.70% to 47.63% along with the participating of more detailed information". It is not a straightforward comparison bewteen the two different paradigms.
>
>     Reffering to the new table above, Line (i) ("BLIP-2", 40.70%) refers basis of the VLM paradigm that directly uses the BLIP-2 for OK-VQA task; Line (h) ("Ours w/o BLIP-2", 44.86%) refers to the basis of the LLM-based paradigm that cooperates VLMs together with LLMs for OK-VQA task. As we integrate the BLIP-2 answers into our "Image Information", we provide the vanilla BLIP-2 accuracy in this table as the starting point. Seen from the new table above, we progressively make used of accessible resources and increase to 47.63% from the basis of the VLM paradigm (Line (i), 40.70%) and the basis of the VLM+LLM paradigm (Line (h), 44.86%) as follows:
>
>     - Line (i) vs Line (h): the comparison of baseline scores for the two paradigms. The VLM paradigm directly uses BLIP-2 for the OK-VQA task (Line (i), 40.70%), and the LLM-based paradigm cooperates VLMs together with LLMs for the OK-VQA task (Line (h), 44.86%);
>     - Line (h) to Line (g): including BLIP-2 answers in the "Image Information" to the vanilla VLM+LLM paradigm in Line (h) (from 44.86% to 45.17%);
>     - Line (g) to Line (f): further adding image information obtained from our generated questions with BLIP-2 answers (from 45.17% to 45.25%);
>     - Line (f) to Line (e): applying refining on the "Image information" (from 45.25% to 47.63%).
>
> - **(8) Explanation of 'n' and '$ \mathcal{E} $*'.**
>     - 'n' refers to the number of shots. It is renamed as "Number of Shots" in the new table above.
>     - '$ \mathcal{E} $*' refers to the format engineering on the few-shot examples, it means to apply filters to refine the image information in the few-shot examples from the training set. We rename it as "Refinement on Few-shot Examples" in the new table above.
>
> **Question 3: The case study**
>
> **Response 3:**
>
> - The reasoning procedure of Prophet involves two stages:
>     - Stage 1: the VQA model produces 10 candidates with confidences;
>     - Stage 2: the LLM predicts the final answer considering the provided candidates.
> - In Figure 3 (1) in our paper, the actual prediction of Prophet is "ham and cheese".
>
>     The "grilled cheese (0.66)" is just one of the intermediate results from "Stage 1" above, while the "ham and cheese" is the final prediction of Prophet in this case. This case implies that even if the VQA model produces correct candidates ("grilled cheese") with reasonable confidence ("0.66"), the LLM may not follow the suggestions when performing reasoning.
>
>     But in our method, we can rectify this wrong LLM prediction and generate "grilled cheese" (prediction of Prophet+ours) instead of "ham and cheese" (prediction of Prophet).
>
> **Question 4: Missing reference**
>
> **Response 4:**
>
> - "Investigating Prompting Techniques for Zero- and Few-Shot Visual Question Answering" is a contemporaneous work to ours, but we'll include it in the revised version.
> - "From Images to Textual Prompts" is already cited and is mentioned as "Img2Prompt" in our paper.

---

### Official Review · Reviewer_Ka9D · 2023-08-06

**Typos Grammar Style And Presentation Improvements:** See the weaknesses section.
**Soundness:** 4

**Excitement:**

4: Strong: This paper deepens the understanding of some phenomenon or lowers the barriers to an existing research direction.

**Missing References:**

Some related work that might be important to look at:
- https://openreview.net/pdf?id=kdHpWogtX6Y  - LLM coordinates information from VLMs.
- https://arxiv.org/abs/1909.04696 - they generate entailed questions to the question asked to improve VQA consistency.

**Paper Topic And Main Contributions:**

Authors design a pipeline that prompts LLMs to ask clarification questions to a vision-language model for the LLM to answer a question accurately about an image. They show that this outperforms simply using the vision-language models for answering and some other similar LLM-based methods on the OK-VQA and AOK-VQA datasets.

**Questions For The Authors:**

- It is unclear what parts are trained and what aren’t. What parts of what models are trained when training the refinement using the VQA annotations? Is the entire pipeline trained end-to-end?
- For the baselines, are the VLM models also trained on the VQA datasets used?

**Reasons To Accept:**

- Interesting design choice requiring very minimal training of interfacing LLM and VLMs, instead of training adapters for image input to llm like llava etc.

- The accuracies on the VQA datasets look promising.

- They show their approach holds with various LLMs.

**Reasons To Reject:**

- The results section could be organized better. Some lines like “Line (f) converts line (h) to few-shot setting” - are lacking details. What does it exactly mean? Fewer context? Fewer VQA training data points? How are they sampled?

- Another baseline to include is the accuracy of the LLM to answer the question without any image information or just the caption. Table 6 compares the accuracy to using BLIP-2 in answering the question. Since their final method has the LLM answering the question, it would be nice to see how much the BLIP and filtered information is helping the LLM.


**Reproducibility:**

3: Could reproduce the results with some difficulty. The settings of parameters are underspecified or subjectively determined; the training/evaluation data are not widely available.

**Reviewer Confidence:**

4: Quite sure. I tried to check the important points carefully. It's unlikely, though conceivable, that I missed something that should affect my ratings.

---

> ### Author Rebuttal · Authors · 2023-08-29
>
> **Question 1: What parts are trained when training the renement module?**
>
> **Response 1:**
>
> - These models are frozen in our entire pipeline: the captioning model ($M_c$), the VQA model ($M_a$) and the LLMs.
>
> - The refinement module in Figure 2 requires training, which includes 4 parts:
>     - An image encoder(for image): to encode the original VQA image;
>     - A question encoder(for text): to encodes the original VQA question;
>     - An information encoder(for text): to encode our generated image information (denoted as "Info encoder" in Figure 2 in our paper);
>     - A filter: an MLP-based network that evaluates the helpfulness of the obtained image information towards answering the visual questions.
>
> **Question 2: Is the entire pipeline trained end-to-end?**
>
> **Response 2:**
>
> No. The LLMs and VLMs in our entire pipelin are frozen. Only the refinement module requires training (as stated above). For the training of the refinement module, we first construct target labels according to Appendix A in our paper. And then we train the encoders and the filter with BCE loss (please refer to Equation 9 on Line 269 in our paper).
>
> **Question 3: For the baselines, are the VLM models also trained on the VQA datasets used?**
>
> **Response 3:**
>
> Yes, Prophet and PromptCap also require training on the VQA datasets, but they train VLMs for different purposes. Prophet trains different VQA models on OK-VQA and A-OKVQA datasets. PromptCap does not involve the training of VQA models, but its image captioning model are trained on the VQA datasets.
>
> **Question 4: Clarification of the results**
>
> **Response 4:**
>
> Sorry for causing the confusion. We carefully revise Table 4 in our paper. Here is the reformed version:
>
> | Method | Number of Shots| Caption | Image Information | Tags | Refinement on Query Instance | Refinement on Few-shot Examples | ACC |
> |---------------------|---|-----------|---------------------------|-----|-----|-----|----------|
> | Ours                | 16| PromptCap | Refined information    |&#10004;|&#10004;|&#10004;|  54.42(a)|
> | Ours                | 16| PromptCap | Refined information       |&#10004;|&#10004;| -   |  52.25(b)|
> | Ours                | 4 | PromptCap | Refined information       |&#10004;|&#10004;| -   |  50.19(c)|
> | Ours                | 4 | OFA-Large | Refined information       |&#10004;|&#10004;| -   |  48.43(d)|
> | Ours                | 4 | OFA-Large | Refined information       | -   |&#10004;| -   |  47.63(e)|
> | Ours w/o refinement | 4 | OFA-Large | All the questions + BLIP-2 answers| -   | -   | -   |  45.25(f)|
> | Ours w/o refinement | 4 | OFA-Large | Original question + BLIP-2 answers| -   | -   | -   |  45.17(g)|
> | Ours w/o BLIP-2     | 4 | OFA-Large | Original question         | -   | -   | -   |  44.86(h)|
> | BLIP-2              | 0 | -         | -                         | -   | -   | -   |  40.70(i)|
>
> Recalling the prompt template for answer reasoning in Section 3.3 in our paper, the template consists of two parts:
>
> - Few-shot in-context examples (Line 287 to Line 288 in our paper), denoting as ***Template-Few-shot***:
>
>         Image information: S^n \n Caption: C^n \n Question: q^n \n Answer: a^n
>
> - A query instance (Line 290 to Line 291 in our paper), denoting as ***Template-Query***:
>
>         Image information: S \n Caption: C \n Question: q \n Answer:
>
> With the in-context examples, the LLM is required to predict the answer for the question q in the query instance.
>
> Here are some explanations of the table above:
>
> - "Caption": the models used to generate the image captions;
> - "Image Information": the resources of our image information, "Refined information" = applying refining on "All the questions + BLIP-2 answers";
> - "Tags": whether to use image tags. According to [PICa](https://arxiv.org/pdf/2109.05014.pdf), image tags are place right after the captions in the template above;
> - "Refinement on Query Instance": applying refining to the "Image information S" in ***Template-Query*** mentioned above;
> - "Refinement on Few-shot Examples": applying refining to the "Image information S^n" in ***Template-Few-shot*** mentioned above;
>
> For the question of **“Line (f) converts Line (h) to few-shot setting”. What does it exactly mean? Fewer context? Fewer VQA training data points? How are they sampled?**
>
> - It should be Line (g) (previous Line (f)) vs Line (i) (previous Line (h)) in this new table. Line (g) and Line (i) belong to different paradigms. Line (i) follows the VLM paradigm that directly applies the VLM (BLIP-2) for OK-VQA task. Line (g) belongs to the LLM-based paradigm where LLMs cooperate with VLMs for OK-VQA task. "Line (g) converts Line (i) to few-shot setting" means that the LLM is making use of BLIP-2 answers for the final reasoning in a few-shot learning way.
> - Line (i) reports the zero-shot BLIP-2 result taken from [the BLIP-2 paper](https://arxiv.org/pdf/2301.12597.pdf). While Line (g) is a 4-shot setting where each query instance is provided with 4 examples (following ***Template-Few-shot***, n=4). These examples are selected according to the relevance of images and questions to the query instance. The BLIP-2 answers are included in the "Image information S^n" in ***Template-Few-shot*** and "Image information S" in ***Template-Query***.
>
> Here are the main conclusions of this table in brief:
>
> - Line (e) vs Line (f): refinement module significantly improves the accuracy (+2.38%);
> - Line (g) vs Line (h): BLIP-2 answers help (a bit) the LLM to reason for answers (+0.31%);
> - Line (d) vs Line (e): tags are helpful (+0.80%);
> - Line (b) vs Line (c): increasing the number of shots significantly improves the accuracy (+2.06%);
> - Line (a) vs Line (b): applying refinement on the few-shot examples markedly boost the accuracy (+2.17%).
>
> These are the main results of the ablation study that claim the importance and necessity of every details in our framework.
>
> **Question 5: How much the BLIP and filtered information is helping the LLM？**
>
> **Response 5:**
>
> - How much the BLIP is helping the LLM?
>
>     We compare the results of with and without BLIP-2 under 4-shot setting. Line 1 in the table below is the baseline reuslt of using just the captions (without "Image Information"). Introducing BLIP-2 improves the accuracy by 1.96%.
>
>     |No.|BLIP-2|Caption|ACC|
>     |:-|:-----|:------|:--|
>     |(1)| No  | OFA-Large |43.21|
>     |(2)| Yes | OFA-Large |45.17 (+1.96)|
>
> - How much the filtered information is helping the LLM?
>
>     Including all the information increases the accuracy by only 0.08% (Line 1 vs Line 2), while using refinement module to produce the filtered information increases the accuracy by 2.46% (Line 1 vs Line 3). The refinement module alone contributes +2.38% to the final performance (Line 2 vs Line 3).
>
>     |No.|Refinement|Image Information|ACC|
>     |:-|:-----|:--------|:--|
>     |(1)| No  |Original question + BLIP-2| 45.17 |
>     |(2)| No  |All the information|45.25|
>     |(3)| Yes |Filtered information| 47.63 |
>
> **Question 6: Missing references**
>
> **Response 6:**
>
> We appreciate for your suggestions and will include them in the Relate Work section. In the first paper, "Language Models are Visual Reasoning Coordicators", the LLM simultanuously reads the output of two VLMs (BLIP and OFA) to predict the answer. In our pipeline, different VLMs are progressively made use of by the LLM. It could be our futher work to build a framework where LLMs automatically decides the paradigm of how to utilize different VLMs for reasoning. The second paper "Sunny and Dark Outside?! Improving Answer Consistency in VQA through Entailed Question Generation" focuses on generating logically consistent question-answer pairs. In our paper, we encourage the LLM to explore different knowledge fields that are somewhat helpful to answering the questions. But the logical consistency is a great insight for us to further improve the generation of new questions.

---

### Official Review · Reviewer_QtV8 · 2023-08-12

**Soundness:** 3

**Excitement:**

3: Ambivalent: It has merits (e.g., it reports state-of-the-art results, the idea is nice), but there are key weaknesses (e.g., it describes incremental work), and it can significantly benefit from another round of revision. However, I won't object to accepting it if my co-reviewers champion it.

**Missing References:**

1. Deyao Zhu, Jun Chen, Kilichbek Haydarov, Xiaoqian Shen, Wenxuan Zhang, Mohamed Elhoseiny. ChatGPT Asks, BLIP-2 Answers: Automatic Questioning Towards Enriched Visual Descriptions. I think this paper should be included in the reference list (this paper was released in ArXiv on 03/12/2023, which is more than three months before the EMNLP deadline). This work also introduces an approach to make LLM and VLM interact to generate more questions so that we can obtain more visual information in the end. After summarization, we can have a caption containing more details about the image.

2. Haoxuan You, Rui Sun, Zhecan Wang, Long Chen, Gengyu Wang, Hammad A. Ayyubi, Kai-Wei Chang, Shih-Fu Chang. IdealGPT: Iteratively Decomposing Vision and Language Reasoning via Large Language Models. This paper is similar to your work to some extent. I hope it can be included in the 'New Question Generation for VQA' section in related work. Since it is contemporaneous work, I just suggest you do so.

**Paper Topic And Main Contributions:**

Paper Topic:
This paper proposes a method to utilize LLM to ask more questions about the image so that LLM can obtain more details about the image to fill the image information gap. After obtaining more visual information, LLM can better handle visual reasoning tasks like OK-VQA and A-OKVQA. In order to generate useful image information, the authors introduce a framework consisting of three modules, i.e., inquiry, refinement, and answering.

Main Contributions:
1. The authors design a model-agnostic framework that can be applied in different LLMs to help them proactively interact with VLMs to unveil missing image information.
2. The proposed method can correct wrong information and minimize the ambiguity of converted textual information.
3. The proposed method can improve baseline models and attain improvements across different LLMs.

**Questions For The Authors:**

Question A: Why not try dense captioning to obtain more image information? (this question has been included in the Reasons To Reject section)

**Reasons To Accept:**

1. The idea of this paper is relatively interesting, proposing a method that leverages the interaction between LLM (Language Large Models) and VLM (Vision-Language Models) to obtain finer-grained image information, which in turn aids visual reasoning tasks. It offers the research community a new perspective on interactive use with large models. Simultaneously, it addresses, to some extent, the current limitation of LLMs being unable to access visual information.

2. The experiments in the article are comprehensive, with rich visualization results. These contents validate the points proposed in the paper. The performance comparisons and ablation studies demonstrate the efficacy of the proposed method. Furthermore, tasks like OKVQA and AOKVQA are inherently challenging. The method presented in this paper achieves competitive results on these tasks, further underscoring its effectiveness.

3. The method put forward in the paper is not only effective but also intuitive, easy to replicate, and clear-cut.

**Reasons To Reject:**

1. Generally speaking, I believe the novelty is somewhat lacking. For instance, earlier this year (published on arxiv on March 12th, more than three months before the EMNLP deadline), there was a paper titled "ChatGPT Asks, BLIP-2 Answers: Automatic Questioning Towards Enriched Visual Descriptions" that employed an approach similar to what you've proposed, aiming to capture more visual information to produce image captions with richer details. Admittedly, in terms of the overall workflow, the structure of your paper seems more comprehensive, encompassing broader considerations. However, on the whole, I don't feel surprised when I read your paper, as the core content bears resemblances to previous work.

2. I believe you should explore one or two additional vision-language tasks. As you've acknowledged in the limitation section, your experiments solely focus on OKVQA and AOKVQA tasks. Essentially, these two tasks are quite similar, with AOKVQA being a more advanced version of OKVQA. Based on the results in Table 3, the performance of OKVQA Prophet (ensemble) only improves by a mere 0.2% with the addition of your proposed method, which hardly convinces me of its efficacy in this context. A 0.2% increment could even be attributed to minor experimental errors, rendering the results somewhat dubious. Hence, rather than tackling both AOKVQA and OKVQA, it would be more sensible to solely concentrate on AOKVQA. On the one hand, the outcomes from Table 3 for AOKVQA appear reasonable and credible; on the other hand, it avoids the redundancy of undertaking two similar tasks. If a wider array of tasks were addressed, incorporating both wouldn't be an issue, but with just these two, their similarity is more glaring. Therefore, if you decide to pursue two tasks, I'd recommend focusing on Visual Commonsense Reasoning (VCR https://visualcommonsense.com/) and AOKVQA. I noticed that you mentioned VCR in the limitation section as well. Personally, I'm more intrigued by the combination of VCR and AOKVQA. I believe readers could glean more insights from these two tasks, given the significant differences in task format (direct generation vs. multiple-choice) and textual content between VCR and AOKVQA. Such diverse content seems far more engaging to me.

3. Given the need to extract more detailed image information for enhanced visual reasoning, why haven't you considered dense captioning? Would using dense captioning yield favorable results? (You can refer to the approach on https://github.com/JialianW/GRiT) Naturally, integrating dense captioning might introduce the problem of hallucination, making the outcomes potentially noisier. Despite the possibility of such issues arising, I'd like to see some experiments incorporating dense captioning, whether in the main content or the appendix.

4. This is a frequently discussed question. Your work stems from the perspective that LLM cannot visualize images, which is indeed correct at present. However, with the evolution of MLLM and the anticipated unveiling of GPT-4's multimodal capabilities, the value of your work is bound to face significant challenges. From my viewpoint, I'd expect authors working on LLM Aided visual reasoning to proactively address the challenges presented by GPT-4 and its implications on their research. However, I haven't observed this aspect in your work.

**Reproducibility:**

4: Could mostly reproduce the results, but there may be some variation because of sample variance or minor variations in their interpretation of the protocol or method.

**Reviewer Confidence:**

4: Quite sure. I tried to check the important points carefully. It's unlikely, though conceivable, that I missed something that should affect my ratings.

---

> ### Author Rebuttal · Authors · 2023-08-29
>
> **Question 1: Differences between our method and ChatCaptioner ("ChatGPT Asks, BLIP-2 Answers: Automatic Questioning Towards Enriched Visual Descriptions")**
>
> **Response 1:**
>
> - We focus on different aspects of image information. ChatCaptioner aims to capture richer details from images as image captions. However, this approach inevitably introduces noise, i.e., inaccurate descriptions of the images. For many vision-and-language (VL) tasks, such as VQA, this noise could lead the model to make incorrect predictions. Therefore, our focus is not only on capturing more image information but also on effectively extracting the useful parts from this information for specific VL tasks.
> - We design a refinement module to successfully select proper information that will have positive effect on answering the questions. As reported in Table 2 and Table 5(a) in our paper, using all the information **without filtering** ("All" in Table 5(a)) results in an accuracy of only 58.03%, while it is 57.91% without our image inforamtion ("Prophet" in Table 2). By introducing our designed refinement module, the accuracy increases to 59.34% ("Prophet+Ours" in Table 2). A similar phenomenon can be observed from the dense caption experiments below (Response 4).
> - We appreciate your suggestions and will include ChatCaptioner in our revised version.
>
> **Question 2: Efficacy compared with Prophet**
>
> **Response 2:**
>
> We would like to further emphasize the improvements of our method compared to Prophet in Table 3 in our paper as follows:
>
> - For A-OKVQA, our single query result (57.9%, "Prophet+Ours" in Table 3) is already comparable to the ensemble result of Prophet (58.2%, "Prophet (ensemble)") with a significantly lower inference cost. Furthermore, our ensemble performance reaches 59.8% ("Prophet+Ours (ensemble)" in Table 3), marking a 1.6% improvement.
>
> - For OK-VQA, we obtain an increment of 1.4% over Prophet in the single query setting ("Prophet+Ours"). In the ensemble setting, we can observe a performance improvement of 0.2% ("Prophet+Ours (ensemble)").
>     - One factor that might contribute to this reduced improvement in the ensemble setting is the inherent characteristics of the Prophet method. Prophet relies on a fine-tuned VQA model to generate answer candidates from a closed-set of predefined answers (all the answers present in the training dataset).
>     - This constraint is particularly noteworthy for OK-VQA, given its lack of answer diversity compared to A-OKVQA. The OK-VQA dataset has 11,508 distinct answer categories, with each question having a maximum of 5 different answers. On the other hand, the A-OKVQA dataset contains 29,631 unique answer categories, with each question having up to 10 distinct answers.
>     - By integrating more few-shot examples, the answer candidates will present a dataset bias. For OK-VQA dataset, we hypothesize that Prophet can benefit from the biased candidates in the ensemble setting.
>
> - We conduct further examination on the ensemble behavior on OK-VQA dataset and find out that Prophet indeed benefits more from ensemble compared to the other baselines. The table below shows the results of the single query setting and the ensemble setting. For Prophet, the gain of the ensemble setting over the single setting is 3.19% (Line 5), which is more than doubled compared to results on Line 1 to Line 4, and is nearly doubled compared to Line 6. This shows an inconsistency regarding the gain of ensemble setting over single setting, and supports our hypothesis that Prophet benefits more from the ensemble setting compared to other methods.
>
>     |No.|    Method     | Single        | Ensemble      | Gain of Ensemble over Single|
>     |:--|:--------------|:--------------|:--------------|:-----------------:|
>     | 1 | PICa          | 49.67         | 51.36         | 1.69             |
>     | 2 | PICa+ours     | 53.76 (+4.09) | 54.91(+3.55)  | 1.15             |
>     | 3 | PromptCap     | 53.50         | 54.16         | 0.66             |
>     | 4 | PromptCap+ours| 54.42 (+0.92) | 55.01 (+0.85) | 0.59             |
>     | 5 | Prophet       | 57.91         | 61.10         | **3.19**         |
>     | 6 | Prophet+ours  | 59.34 (+1.43) | 61.30 (+0.20) | 1.96             |
>
> - Therefore, we consider the reduced improvement in the OK-VQA + ensemble setting to be a special case, whereas in most other cases our approach leads to significant improvements compared to Prophet.
>
> **Question 3: Experiments on new tasks that are more different from OK-VQA and A-OKVQA**
>
> **Response 3:**
>
> - Exploring other VL tasks, such as VCR, is a highly insightful suggestion. We are now working on it. However, due to the extensive transfer work required, we regret that we were unable to provide the results within the given time constraints. We will strive to include the relevant findings in subsequent versions.
>
> **Question 4: Dense caption experiments**
>
> **Response 4:**
>
> - In our framework, the LLM perceives the image from two parts of the prompt: "Caption" and "Image Information" (as in the template shown between Line 288-289 in our paper). We incorporate the dense captions (denoted as GRiT) and conduct the following two types of experiments with the PICa pipeline:
>     - dense captions directly as "Caption" (Line 2);
>     - dense captions as a type of "Image Information" (Lines 4 and Line 5).
>
>     |No.|                   Method   | Caption | Image Information | ACC   |
>     |:-|:----------------------------|:--------|:------------------|:------|
>     |1 | PICa                        | OSCAR   | -                 | 49.67 |
>     |2 | PICa                        | **GRiT**| -                 | 46.16 (-3.51) |
>     ||
>     |3 | PICa + Ours                 | OSCAR   | Ours              | 53.76 |
>     |4 | PICa + Dense Caption        | OSCAR   | **GRiT**          | 49.43 (-4.33) |
>     |5 | PICa + Ours + Dense Caption | OSCAR   | Ours + **GRiT**   | 52.55 (-1.21) |
>
> - In the table above, Line 1 and Line 2 show the influence of dense captions and general captions. Simply replacing the OSCAR captions by GRiT captions reduces the accuracy from 49.67% to 46.16% (-3.51%). Adding GRiT to PICa as the "Image Information" (Line 4) also impairs the original PICa performance by 0.24% (comparing Line 4 to line 1).
> - Lines 3, Line 4 and Line 5 compare the performances of different types of "Image Information". Comparing Line 3 and Line 4, replacing our information by GRiT's dense captions decreases the performance by 4.33%. While combining Ours and GRiT captions as the "Image Information" reduces the accuracy by 1.21% compared to using only Ours "Image Information" (comparing Line 5 to Line 3).
> - To conclude, dense captions introduce noise and degrade accuracy, regardless of their role as either "Caption" or "Image Information". In contrast, utilizing the image information obtained and refined by our method consistently yields the best results.

---

### Meta-Review · Area_Chair_2TEo · 2023-09-19

**Recommendation:** 3

**Metareview:**

This paper introduces a framework that enables LLMs to proactively ask relevant questions to unveil more details in the image and evaluate the method on two visual QA datasets, OK-VQA and A-OKVQA. I think the idea explored in this paper is interesting, the performance gain given the complexity of additional modules seems somewhat limited, as pointed out by the reviewers. I am also curious whether this method can give further improvements on top of existing models on other tasks, as suggested by the reviewer QtV8

---

### Decision · Program_Chairs · 2023-10-07

**Decision:**

Accept-Findings

**Comment:**

This paper introduces a framework that enables LLMs to proactively ask relevant questions to unveil more details in the image and evaluate the method on two visual QA datasets, OK-VQA and A-OKVQA. I think the idea explored in this paper is interesting, the performance gain given the complexity of additional modules seems somewhat limited, as pointed out by the reviewers. I am also curious whether this method can give further improvements on top of existing models on other tasks, as suggested by the reviewer QtV8